# Inhibition of Metastatic Hepatocarcinoma by Combined Chemotherapy with Silencing VEGF/VEGFR2 Genes through a GalNAc-Modified Integrated Therapeutic System

**DOI:** 10.3390/molecules27072082

**Published:** 2022-03-24

**Authors:** Xunan Li, Xiang Wang, Nian Liu, Qiuyu Wang, Jing Hu

**Affiliations:** 1Wuxi School of Medicine, Jiangnan University, Lihu Avenue 1800, Wuxi 214122, China; lixunan3022@163.com (X.L.); lnxy713@163.com (N.L.); qyw9812@163.com (Q.W.); 2Key Laboratory of Carbohydrate Chemistry and Biotechnology, Ministry of Education, School of Biotechnology, Jiangnan University, Wuxi 214122, China; wx1111123456@163.com

**Keywords:** hepatocellular carcinoma, metastasis, ASGPR, gene therapy, VEGF/VEGFR2 signaling pathway

## Abstract

Hepatocellular carcinoma (HCC) is a highly malignant tumor related to high mortality and is still lacking a satisfactory cure. Tumor metastasis is currently a major challenge of cancer treatment, which is highly related to angiogenesis. The vascular endothelial growth factor (VEGF)/VEGFR signaling pathway is thus becoming an attractive therapeutic target. Moreover, chemotherapy combined with gene therapy shows great synergistic potential in cancer treatment with the promise of nanomaterials. In this work, a formulation containing 5-FU and siRNA against the VEGF/VEGFR signaling pathway into N-acetyl-galactosamine (GalNAc)-modified nanocarriers is established. The targeting ability, biocompatibility and pH-responsive degradation capacity ensure the efficient transport of therapeutics by the formulation of 5-FU/siRNA@GalNAc-pDMA to HCC cells. The nano-construct integrated with gene/chemotherapy exhibits significant anti-metastatic HCC activity against C5WN1 liver cancer cells with tumorigenicity and pulmonary metastasis in the C5WN1-induced tumor-bearing mouse model with a tumor inhibition rate of 96%, which is promising for future metastatic HCC treatment.

## 1. Introduction

Hepatocellular carcinoma (HCC) is a highly heterogeneous and malignant tumor with a poor prognosis. It remains the sixth most prevalent malignancy and the fourth leading cause of cancer-related mortality worldwide [1]. At early stages, HCC patients can usually receive liver transplantation or surgical resections as a curative treatment [2]. However, most patients are identified as advanced HCC at the time of diagnosis [3,4]. Despite remarkable progress in earlier diagnosis and novel therapeutic agents in the past decades, the 5 year survival rate of HCC patients remains remarkably poor due to serious extrahepatic metastases and a high recurrence rate [5,6]. To deal with this issue, nanomaterial-based delivery platforms attracted great attention to exploit multimodal therapies to achieve synergistic anti-tumor efficacies, such as liposomes [7], micelles [8], protein nanoparticles [9], carbon-based materials [10,11] and metal and covalent organic frameworks [12,13].

Tumor metastasis, the dissemination of primary tumor cells from the initial site to distant sites to form secondary tumors, is currently a major challenge in cancer treatment [14,15]. In many cases, metastases are already developed when primary cancers are diagnosed [16]. Cancer cells can escape to other tissues or organs through the blood vessels, and tumor vessels supply tumor cells with nutrients and oxygen to support their continued growth, invasion and metastasis [17,18,19]. The inhibition of tumor angiogenesis to block the tumor’s blood supply has become an effective anti-tumor growth and metastasis strategy in cancer treatment [20]. The vascular endothelial growth factor (VEGF) is reported to be more expressed in liver cancer tissues than in normal liver tissues and para-neoplastic tissues, and the VEGF/VEGFR2 signal transduction pathway plays a major role in peritumoral angiogenesis, which is involved in the migration, proliferation and survival of perivascular new endothelial cells. It becomes a promising alternative to block the VEGF/VEGFR2 signaling pathway to inhibit the generation and migration of solid tumors [21]. 

Recently, the RNA interference (RNAi) system based on gene silencing mechanisms is utilized as a powerful tool in anti-tumor gene therapy [22,23]. Small interfering RNAs (siRNAs) have the capability to regulate the expression of RNA transcripts, induce mRNA degradation or repress protein translation [24]. To deliver the negatively charged naked siRNA, some challenges need to be faced during systemic circulation, including rapid metabolism, off-target effects and RNase degradation [25,26]. In view of this fact, to maximize the gene silencing efficiency of siRNA, it is highly needed to utilize a stable delivery system to make siRNAs effectively and accurately recognize and reach the targeted tumor sites after long-term circulation [26].

Herein, we synthesized an integrated delivery system based on two monomers [3-azido-2-hydroxypropyl methacrylate (AHPMA) and 2-(dimethylamino) ethyl methacrylate (DMAEMA)] to combine gene therapy against VEGF and VEGFR2 and chemotherapy to inhibit hepatocarcinogenesis and metastasis, which can be initiated with a cancer stem cell (CSC)-like cell line C5WN1 with tumorigenicity and pulmonary metastasis (Figure 1) [27]. The monomer DMAEMA can form a stable complex with amino groups of negatively charged nucleic acids to prevent enzymatic degradation. The sponge effect triggered by acidic pH value can cause polymeric DMAEMA swelling and endosomal escape, thus achieving gene transfection [28,29]. These features make this platform suitable for nucleic acid concentration and transport. 5-Fluorouracil (5-FU), a commonly used anti-cancer reagent with an inhibitory effect on cell growth and migration, is encapsulated into the formulation for chemotherapy. Asialoglycoprotein receptor (ASGPR) is a mammalian lectin specifically expressed on the surface of hepatocytes and is an attractive hepatic delivery target [30,31]. GalNAc residues can mediate the specific cellular uptake of the nanoparticle by targeting ASGPR expressed liver cells as one of the specific ligands for ASGPR. The self-assembled nanoparticles are multiple-functionalized by modifying them with N-acetyl-galactosamine (GalNAc) for hepatic targeted delivery and rhodamine B (RhB) for fluorescein-based tracking. GalNAc residues can be linked to the azido group in AHPMA through click reaction. 

## 2. Results and Discussion

### 2.1. Synthesis and Characterization of 5-FU/siRNA @GlaNAc-pDMA

The GalNAc-modified nanocarrier with fluorescence labeling was constructed. GalNAc [32] (Appendix A), RhB-based atom transfer radical polymerization (ATRP) initiator (RhB-Br) [33] (Appendix A) and monomeric AHPMA [34,35] (Appendix A) were first synthesized using previously reported methods. The fluorescently labeled and pH-responsive block copolymer p(RhB-DMAEMA-AHPMA-GalNAc) was prepared based on atom transfer radical polymerization (ATRP) with RhB-Br and AHPMA, which was verified by ^1^H-NMR spectroscopy (Appendix A). Additionally, the copolymer p(RhB-DMAEMA-AHPMA-GalNAc) were obtained by a further “click” reaction with GalNAc. All proton signals of p(RhB-DMAEMA-AHPMA-GalNAc) were clearly shown in ^1^H-NMR spectra (Appendix A), representing the successful synthesis of p(RhB-DMAEMA-AHPMA-GalNAc). The results of Fourier transform infrared (FT-IR) spectroscopy also confirmed the correct construction (Appendix A).

The glyco-copolymers p(RhB-DMAEMA-AHPMA-GalNAc) was then self-assembled into nanoparticles (GalNAc-pDMA) via a solvent exchange method. Transmission electron microscopy (TEM) micrographs showed that the assembled particles are roughly spherical in shape and with relatively homogeneous size (Figure 1A and Appendix A). Meanwhile, the size of GalNAc-pDMA particles was confirmed by dynamic light scattering (DLS) with an average diameter of 198.1 ± 0.4 nm and a polydispersity index (PDI) value of 0.24 ± 0.085 (Figure 1B). Hydrophobic 5-FU was trapped into the hydrophobic cores of GalNAc-pDMA by dialysis, which was confirmed with UV-spectroscopy (Appendix A). A simple dialysis method was applied to trap hydrophobic 5-FU into the hydrophobic cores of GalNAc-pDMA. The drug loading content (DLC) and drug entrapment efficiency (DEE) of GalNAc-pDMA were determined by measuring 5-FU absorption at 265 nm to be 12.6 ± 0.03% and 58.0 ± 3.4%, respectively (Appendix A).

The cationic DMAEMA unit can form complexes with negatively charged siRNA fragments. The siRNA condensing capacity of GalNAc-pDMA was determined by the nitrogen/phosphate ratio [N/P, nitrogen (N) of the PMAEMA moiety in GalNAc-pDMA and phosphate (P) in RNA] from 0.5 to 10, and naked siRNA was used as a negative control. The results of the agarose gel retardation assay showed that GalNAc-pDMA form complexes with a mixture of VEGF-siRNA andVEGFR2-siRNA with Cy5 labelling (sequence information see Appendix A) at an N/P ratio of 9:1 (Figure 1F). After siRNA was loaded, the zeta potential reduced from 19.5 to −2.5 mV (Figure 1C), mostly due to the negative charge of siRNA, also confirming the siRNA condensing capability of GalNAc-pDMA.

To evaluate the stability of 5-FU/siRNA@GalNAc-pDMA, the hydrodynamic diameter and zeta potential of 5-FU/siRNA@GalNAc-pDMA in different solutions such as water, PBS (pH = 7.0) and cell culture medium (DMEM containing 10% fetal bovine serum (FBS) were measured for seven consecutive days (Figure 1D). The result showed that the particles remained relatively stable under physiological conditions. 

The drug release of 5-FU/siRNA@GalNAc-pDMA was further measured under different pH conditions at 37 °C, mimicking the physiologic status (pH 7.4) and the periphery of the tumor (pH 5.0). The release profile of 5-FU showed that only 18% of 5-FU was released after 72 h in PBS at pH 7.4. Comparably, nearly 80% of 5-FU was released within 12 h under the acidic pH conditions (pH 5.0) (Figure 1E). It demonstrated that 5-FU/siRNA@GalNAc-pDMA has great pH-responsive drug release capacity, which thereby can reduce the occurrence of side reactions in normal tissues.

### 2.2. In Vitro Targeted Synergistic Effect of 5-FU/siRNA@GalNAc-pDMA

#### 2.2.1. In Vitro Biosafety and Cytotoxicity Assessment

Good bioactivity and biocompatibility are essential characteristics for exogenous nanomaterials as drug delivery systems. The biocompatibility was assessed via MTT assays and hemolysis evaluation. C5WN1, HepG2 and Huh7 cells were tested, and HEK293 cells were selected as non-cancer cells incubating for 48 h with GalNAc-pDMA (0–600 μg mL^−1^). The MTT assay results showed that viability of cells was greater than 90% in all cell lines even at the highest concentration of GalNAc-pDMA, indicating very low cytotoxicity of the nanomaterial (Appendix A). In addition, GalNAc-pDMA particles exhibited a very low hemolysis rate even at the highest concentration (<10%) compared to the serious hemolysis caused by water as a positive control with a hemolysis rate of 100% (Appendix A). It suggests that GalNAc-pDMA nanoparticles are suitable as delivery systems with high biocompatibility and low cytotoxicity.

#### 2.2.2. ASGPR-Targeted Intracellular Uptake

The ASGPR-targeting capacity of GalNAc-pDMA particles as a co-delivery system was verified by observation with confocal laser scanning microscopy (CLSM) (Figure 2A) and flow cytometry analysis (Figure 2B). Hepatocarcinoma cells with high ASGPR expression on the cell membrane, such as C5WN1, HepG2 and Huh7, were used for the tests [36,37]. HEK293 cells with low ASGPR expression was used as a negative control. After 3 h incubation with 5-FU/siRNA@GalNAc-pDMA, the cellular uptake behavior of the nanoparticles was detected by CLSM and flow cytometry analysis. Compared with HEK293 cells, significant RhB fluorescence and cy5 fluorescence were both observed on the cell membrane and in the cytoplasm of C5WN1, HepG2 and Huh7 cells, demonstrating the active targeting of 5-FU/siRNA@GalNAc-pDMA towards high ASGPR expressing cells. In addition, there was no obvious difference in detected fluorescence when HEK293 cells were treated with a free galactose-supplemented medium and then incubated with 5-FU/siRNA@GalNAc-pDMA. In contrast, the fluorescence intensity in C5WN1, HepG2 and Huh7 cells became weaker after preincubation with free galactose, as free galactose competed with GalNAc-modified nanoparticles for binding to ASGPR on the cell membrane and decreased the subsequent endocytosis. All these results showed that 5-FU/siRNA@GalNAc-pDMA could effectively target HCC cell lines through the specific recognition of modified GalNAc residues with ASGPR.

#### 2.2.3. In Vitro Therapeutic Effect of Codelivery of 5-FU and siRNA

The in vitro therapeutic effect of 5-FU/siRNA@GalNAc-pDMA was investigated using an MTT assay (Figure 3). A series of concentrations of 5-FU, 5-FU@GalNAc-pDMA and 5-FU/siRNA@GalNAc-pDMA were cultured with C5WN1, Huh7 and HepG2 cells, respectively, in DMEM medium for 48 h. Comparably, the cell inhibitory effect of 5-FU/siRNA@GalNAc-pDMA was significantly more enhanced than those of free 5-FU and 5-FU/siRNA at the 5-FU concentration of 50 μg mL^−1^ (Appendix A), which was then used for the subsequent in vitro experiments. Compared with the control group treated with PBS, the cell viability of the free 5-FU treated group and the free siRNA treated groups was around 70%. The HepG2, Huh7 and C5WN1 cell growth of the encapsulated 5-FU group and encapsulated siRNA group, was greatly inhibited; however, there was no significant difference for HEK293 cells, suggesting the targeted delivery system efficiently increased the cytotoxicity of free 5-FU and siRNA towards the hepatocarcinoma cells. A glucose-modified pDMA (Glc-pDMA) was co-loaded with 5-FU and siRNA to be used as a control. The 5-FU/siRNA@Glc-pDMA treatment did not show apparent synergistic cytotoxicity on all cell lines. The 5-FU/siRNA@GalNAc-pDMA treated groups of HepG2, Huh7 and C5WN1 cells displayed significant cell death with cell viabilities of 32%, 36% and 24%, respectively, while there was no effect on HEK293 cells. These results indicate that 5-FU/siRNA@GalNAc-pDMA has significant superiority for targeted synergistic chemotherapy/gene therapy towards hepatocarcinoma cells.

#### 2.2.4. Inhibition of C5WN1 Migration by Gene Therapy of 5-FU/siRNA@GalNAc-pDMA

It was shown that the VEGF/VEGFR2 signaling pathway plays a crucial role in regulating cell viability, proliferation and migration. The migration ability of HCC cells was also confirmed to be closely related to VEGF and VEGFR expression. In this work, co-loaded siVEGF and siVEGFR were utilized to inhibit the viability and migration of C5WN1 by interfering with the transcriptional process of both genes, thereby downregulating the expression of VEGF and VEGFR2.

To investigate the efficacy of gene silencing, the expression of VEGF and VEGFR2 protein expression levels were evaluated by Western blot. The protein expression levels of VEGF and VEGFR2 were downregulated to 68% and 70%, respectively, after incubation of C5WN1 cells with its respective siRNA alone (Figure 4A). After siVEGF and siVEGFR2 were, respectively, encapsulated with GalNAc-pDMA, the protein expression levels of VEGF and VEGFR2 were further downregulated to 38% and 44%, respectively. The expression of VEGF and VEGFR2 genes was further downregulated, respectively, after the simultaneous encapsulation of siRNA into GalNAc-pDMA particles. Surprisingly, the group treated with 5-FU/siRNA@GalNAc-pDMA showed the lowest VEGF and VEGFR2 protein levels of 28% and 27%, respectively, which suggested that the bimodal therapies of gene/chemotherapy by GalNAc-pDMA delivery system could effectively inhibit the VEGF/VEGFR signaling pathway, thus inhibiting HCC cells migration.

The effect of 5-FU/siRNA@GalNAc-pDMA on the migration of C5WN1 cells was further studied. The cell migration of C5WN1 cells was detected by transwell assay after incubation with different formulations. The numbers of migrated cells of groups treated with encapsulated siRNA were much lower than those of groups treated with naked siRNA (Figure 4B), indicating the efficacy of targeted delivery. The group treated with 5-FU/siRNA@GalNAc-pDMA showed the highest inhibition on cell migration, suggesting the synergistic effect of this construct on the migration of C5WN1 cells.

### 2.3. In Vivo Targeted Synergistic Effect of 5-FU/siRNA@GalNAc-pDMA

In order to further evaluate the in vivo targeted synergistic effect, a C5WN1 subcutaneous tumor model was established [27]. Animal experiments were carried out with the approval of the experimental animal ethics committee of Jiangnan University (Approval Number: JN. No20210415c1350920). The in vivo biodistribution of GalNAc-pDMA was first evaluated. 5-FU@GalNAc-pDMA was injected into mice through the tail vein to observe the fluorescence signal of RhB in mice after 4, 8, 12, 24 and 48 h, and 5-FU@Glc-Micel was used as a non-targeted control group. After 4 h of injection, the nanoparticles were detected to be distributed throughout the whole body in both the targeted group and the non-targeted group (Figure 5A). After 8 h of injection, it was detected that 5-FU@GalNAc-pDMA started to accumulate in the tumor site. At 12 h post-injection, a significant accumulation of 5-FU@GalNAc-pDMA was detected in the tumor site. There was no obvious tumorous accumulation in the non-targeted group compared with the targeted group, indicating the excellent in vivo targeting capacity of 5-FU@GalNAc-pDMA mediated by specific ASGPR recognition. At 24 h after injection, a certain fluorescence intensity in the tumor site was still detected in the targeted group, suggesting prolonged drug retention and effectiveness achieved by the targeted delivery. After 48 h, the systemic fluorescence became very weak in both groups, showing that the nanoparticles could be systemically metabolized. Major organs of the heart, liver, spleen, lung, kidney and tumors were isolated from mice for imaging. Although there was also some accumulation in the tumor of the non-targeted group, mostly due to enhanced permeability and retention (EPR) effect, the fluorescence intensities in tumors of the targeted group were much stronger than those of the non-targeted group (Figure 5B). These results confirmed that 5-FU@GalNAc-pDMA can effectively accumulate in tumor tissues in the long term due to its good stability and targeting ability.

The synergistic therapeutic effect of gene/chemotherapy carried by 5-FU/siRNA@GalNAc-pDMA was then further evaluated with the subcutaneous C5WN1 tumor-bearing mice, which is a CSC-like cell line with tumorigenicity and pulmonary metastasis. Nine groups of mice were randomly separated and intravenously injected with different formulations every two days during two weeks, including saline, GalNAc-pDMA, siRNA (siVEGF+siVEGFR2), Free 5-FU, 5-FU+siRNA, siRNA@GalNAc-pDMA, 5-FU@GalNAc-pDMA, 5-FU/siRNA@Glc-pDMA and 5-FU/siRNA@GalNAc-pDMA, respectively. The tumor volume and bodyweight of mice were checked daily during the entire therapeutic period. On day 14, the tumor inhibition rates (TIR) were calculated based on the final tumor volume using saline as negative control (Figure 6A,B and Appendix A). Compared with the saline control group, the blank micelles had almost no tumor inhibition effect. The free 5-FU and naked siRNA had weak tumor inhibition effects with the TIR of 32% and 33%, respectively. The TIR of encapsulated drug group and encapsulated gene group increased to 57% and 67%, respectively, showing the improved therapeutic effect of targeted delivery compared to individual chemotherapy or gene therapy. The 5-FU/siRNA@GalNAc-pDMA group reached the maximum tumor inhibition among all groups with the TIR of 96%. The tumors and major organs of all mice were dissected for a further check. The tumor mass of 5-FU/siRNA@GalNAc-pDMA group was found to have the lowest weight (Figure 6C), which was in good agreement with tumor volume measurement. In this study, a low intravenous injection dose of 5-FU at 5 mg kg^−1^ was selected in consideration of the low usage amount of nanoplatforms, compared with the reported 5-FU dose at 10–15 mg kg^−1^ in mice [38,39,40]. The significantly enhanced anti-tumor activity of this formulation with bimodal therapies at a low dose of 5-FU demonstrated the synergistic effect of combined gene therapy. The haematoxylin and eosin (H&E) sections of tumor tissues from different groups demonstrated that the most severe cell necrosis occurred in the 5-FU/siRNA@GalNAc-pDMA treated group, and different levels of cell necrosis were observed in other treatment groups, while the tumor cells in the saline and GalNAc-pMDA groups retained intact cell morphology (Appendix A), which supported the previous results. The above results indicated that the GalNAc-pDMA-based formulation carried synergetic gene/chemotherapy and elicited a potent therapeutic effect on metastatic HCC. The subcutaneous tumor model is not sufficient to study the effect of this formulation on extrahepatic metastasis; the inhibition efficacy on HCC metastasis still needs to be further studied in a tumor orthotopic transplantation model.

Moreover, the 5-FU/siRNA@GalNAc-pDMA-treated mice showed negligible body weight loss during the treatment, indicating particularly low systemic toxicity of the formulation (Figure 6D). The H&E sections of the main organs from each experimental group also showed no obvious lesions of the cell morphology, confirming the in vivo biosafety of 5-FU/siRNA@GalNAc-pDMA (Appendix A).

## 3. Materials and Methods

### 3.1. Materials

All chemicals and reagents were commercially available without the need for further purification, except 2-(Dimethylamino) ethyl methacrylate (DMAEMA, 98%, from Sigma Aldrich, Germany), which was distilled at reduced pressure just prior to use. VEGF siRNA and VEGFR2 siRNA were used as reported in the literature (the sequence and primer were listed in Appendix A) and synthesized by Sangon Biotech, Shanghai, China. All of the aqueous solutions used in experiments were prepared using deionized water.

### 3.2. In Vitro Cytotoxicity Evaluation of GalNAc-pDMA

The HCC cell lines HepG2 and Huh7 were obtained from the Chinese Academy of Sciences Cell Bank. HEK293 cells were obtained from the Conservation Genetics CAS Kunming Cell Bank. C5WN1 cells were self-established [27]. C5WN1, Huh7, HepG2 and HEK293 cells were maintained in DMEM medium supplemented with 10% FBS, 1% penicillin and 1% streptomycin (growth medium), in 5% CO_2_ and 95% air at 37 °C. The MTT assay was carried out to determine in vitro cytotoxicity of GalNAc-pDMA. Cells were seeded at a density of 5 × 10^3^ cells per well of 96-well plates in triplicate the day before GalNAc-pDMA treatment. To observe the cellular uptake of GalNAc-pDMA, the cells were incubated with 200 μL of fresh medium containing different formulations at a final GalNAc-pDMA concentration of 60–600 μg mL^−1^ and cultured for 48 h. The control group was treated with an equal volume of culture medium. Then, the medium was removed, and the cells were washed with PBS (pH 7.4) three times. In total, 100 μL of MTT solution (0.5 mg mL^−1^ PBS) were added to each well and kept in the dark for 4 h until purple formazan crystals formed. Finally, the MTT solution was aspirated gently, and 100 μL of DMSO were added. The absorbance of each well was measured at a wavelength of 470 nm with a microplate reader (Bio Tek, USA). Cell viability was calculated using the following equation:Cell Relative viability (%) = (OD_Treated_ − OD_Blank_)/(OD_Control_ − OD_Blank_) × 100%.

### 3.3. Western Blotting Analysis

The VEGF and VEGFR2 expression of samples were tested by Western blotting analysis. C5WN1 cells were subject to different treatments for 48 h incubation. The total protein lysates were obtained after C5WN1 cells were lysed and centrifuged at 12,000× *g* at 4 °C for 15 min. Protein concentrations were measured by BCA protein assay (Biyotime). The same amounts of protein from various treatments were electrophoresed and resolved on 12% SDS-PAGE gels. After protein transferring onto PVDF membranes, the membranes were incubated with specific primary antibodies overnight including a rabbit polyclonal anti-VEGF antibody (Cat. ab46154, Abcam), a rabbit polyclonal anti-VEGFR2 antibody (Cat. ab256666, Abcam) and a rabbit polyclonal anti-β-actin anti-body (Cat. 20536-1-AP, Proteintech). All primary antibodies were incubated in PBS-Tween (PBS-T) and incubated overnight at 4 °C.

The VEGF and VEGFR2 expression of samples were tested by Western blotting analysis. C5WN1 cells were subject to different treatments for 48 h incubation. The total protein lysates were obtained after C5WN1 cells were lysed and centrifuged at 12,000×*g* at 4 °C for 15 min. Protein concentrations were measured by BCA protein assay (Biyotime). The same amounts of protein from various treatments were electrophoresed and resolved on 12% SDS-PAGE gels. After protein transferring onto PVDF membranes, the membranes were incubated with specific primary antibodies overnight including a rabbit polyclonal anti-VEGF antibody (Cat. ab46154, Abcam), a rabbit polyclonal anti-VEGFR2 antibody (Cat. ab256666, Abcam) and a rabbit polyclonal anti-β-actin anti-body (Cat. 20536-1-AP, Proteintech). All primary antibodies were incubated in PBS-Tween (PBS-T) and incubated overnight at 4 °C.

### 3.4. Transwell Assay

C5WN1 cells were incubated in the medium containing different formulations for 12 h. After trypsin digestion, the cells were adjusted to a concentration of 1 × 10^6^ cells/mL and seeded on the upper chamber of Matrigel-coated transwells supplemented with 10% serum DMEM as a chemoattractant. Plates were harvested after 12 h incubation. The transwell membranes were fixed with methanol for 15 min and stained with 1% crystal violet for 10 min. Membranes were cleaned and then mounted on glass slides to record the number of invading cells under a microscope.

### 3.5. Intracellular Uptake Behavior

Intracellular uptake of 5-FU/siRNA@GalNAc-pDMA in hepatoma cells was observed by confocal laser scanning microscope (CLSM) and flow cytometry. For the CLSM assay, HepG2, Huh7 and C5WN1 cells were tested for their high ASGPR expression on the cell membrane; HEK293 cells are human embryonic kidney cells and were used as negative controls in this experiment due to the low expression of ASGPR. Cells (5 × 10^4^ per dish) in DMEM medium supplemented with 10% FBS were seeded onto 35 mm glass-bottom Petri dishes and allowed to grow at 37 °C with 5% CO_2_. After 24 h, the medium was removed, and cells were washed with PBS (pH 7.4) three times. 5-FU/siRNA@GalNAc-pDMA (10 μg mL^−1^) in 1 mL of fresh medium were added and incubated for 4 h. Then, the fluorescence images of samples were acquired by CLSM (Wetzlar, Germany). The fluorescence of cell nuclei, RhB, and Cy5-siRNA was obtained using laser lines at 360, 560 and 650 nm, respectively.

For the flow cytometry assay, C5WN1, Huh7, HepG2 and HEK293 cells were seeded onto 24-well plates at a density of 1 × 10^4^ cells per well. After 24 h incubation with 5% CO_2_ at 37 °C, the cells were incubated in a medium containing siRNA@GalNAc-PDMA (10 or 20 μg mL^−1^) for another 6 h. The cells were then measured by flow cytometry (Franklin, San Mateo, CA, USA). The experiment was repeated at least three times.

### 3.6. In Vivo Biodistribution

When the subcutaneous tumor reached 200 mm^3^, 10 BALB/c nude mice were randomly divided into two groups and were i.v. injected with GalNAc-pDMAor Glc-pDMA (100 µL, 6 mg mL^−1^) at 4, 8, 12, 24 and 48 h. The fluorescence signals of RhB in the anesthetized mice (using CO_2_ asphyxiation) and organs were imagined by Bruker In Vivo Xtreme II (Woltham, MA, USA).

### 3.7. In Vivo Antitumor Effect

Tumor-bearing mice were randomly divided into nine groups (five mice per group) treated with: (1) saline, (2) GalNAc-pDMA, (3) siRNA(siVEGF+siVEGFR2), (4) 5-FU, (5) 5-FU+siRNA, (6) siRNA@GalNAc-pDMA, (7) 5-Fu@ GalNAc-pDMA, (8) 5-FU/siRNA@Glc-pDMA and (9) 5-Fu/siRNA@GalNAc-pDMA. In total, 200 µL of each formulation were injected via the tail vein on days 1, 3, 5, 7, 9, 11 and 13. Tumor sizes and mice body weights were recorded every day. On day 14, tumor tissues and major organs (liver, spleen, kidney, heart and lungs) were acquired for H&E staining. The stained specimens were examined by digital optical microscopy (Bruker, USA).

The tumor inhibition rates were defined as per the following formula:TIR = 1 − (average volume of tumors in experimental group/average volume of saline tumors) × 100%.

### 3.8. Statistical Analysis

The experiments were repeated at least three times. The statistical significance of data was determined by one-way analysis of variance (ANOVA) with Origin Lab. Results were shown as mean ± standard deviation (mean ± SD) (*n* ≥ 3; for animal tests *n* > 6).

## 4. Conclusions

In summary, a comprehensive therapeutic drug delivery system was designed and prepared. The platform was verified to be biocompatible and safe in vitro and in vivo. Modification of GalNAc residues enabled nanocarriers to efficiently and specifically target ASGPR expressed hepatocytes to increase the efficiency of therapeutics and reduce systemic toxicity in vivo. The fabricated nanoplatform achieved an elongated hepatic drug concentration by the stable transport of drug and therapeutic nucleic acids and pH-controlled drug release. The formulation of 5-FU and siRNA against the VEGF/VEGFR signaling pathway exhibited potent synergistic in vivo anti-tumor efficacy on metastatic HCC at a low dose of 5-FU. The strategy of the combined chemotherapy and gene therapy against cell migration has great potential to improve metastatic HCC treatment in future clinical applications.

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
