# Peer review of "Inhibition of Metastatic Hepatocarcinoma by Combined Chemotherapy with Silencing VEGF/VEGFR2 Genes through a GalNAc-Modified Integrated Therapeutic System"

_molecules, 2022, doi:10.3390/molecules27072082_

Round 1
Reviewer 1 Report
This manuscript describes a formulation containing 5-FU and siRNA against VEGF/VEGFR2 genes into a N-acetyl-galactosamine (GalNAc)-modified nanocarriers as anticancer treatment in hepatocellular carcinoma (HCC). The Authors present the synthesis of components and assembly of glyco-copolymers p(RhB-DMAEMA-AHPMA-GalNAc) into nanoparticles (GalNAc-pDMA) and show in vitro and in vivo experiments aimed at demonstrating the anticancer efficacy of this combined chemotherapy against HCC cell lines. The strategy of combining chemotherapy with gene therapy is interesting and original, however the following points should be addressed in order to render the manuscript acceptable for publication.
- The description of the in vitro and in vivo experiments lacks some essential information. In particular, the doses of formulations or their single components (5-FU, siRNA, etc.) are not reported in figures 4 and 6; in figure 3, only the concentrations of 5-FU (50 ug/ml) and GalNAc-pDMA (300 ug/ml) are reported in the caption and the others are missing. The concentrations of each sample, formulation or individual component, must be clearly reported for all experiments. Similarly, the treatment time is not reported for the experiment shown in figure 4A.
- How were the individual doses of formulations or single components used for the in vitro and in vivo experiments chosen? Have dose-response experiments been performed to explore a broad range of efficacy/toxicity, and ultimately to delineate the therapeutic window of these formulations? The Authors should comment on this point, since the therapeutic index is an essential factor in the evaluation of combination therapies.
- The nanoparticle targeting strategy is poorly described in the Introduction and in the Scheme 1. It is necessary to better explain that targeting occurs via binding of GlcNAc to ASPGR expressed by HCC cells and that this confers selectivity with respect to normal liver tissue. Moreover, the acronym ASGPR (asialoglycoprotein receptors) should be spelled out somewhere in the text.
- Paragraph 2.2.1 and figure S10. The text states that water was used as a positive control of hemolysis, but this data is not shown in Figure S10.
- Paragraph 2.2.2. The Authors state that the C5WN1, HepG2 and Huh7 cell lines express high levels of ASGPR on the cell membrane, while the HEK293 line used as a negative control expresses low levels of ASPGR. This claim is not supported by any data shown in this manuscript, therefore it is necessary to demonstrate different levels of ASPGR expression by immunofluorescence, western blot or cytofluorimetry.
- Page 6, figure 3. Why is the PBS-treated control group (normalized to 100%) with its standard deviation bars not shown in the four graphs? What is the statistical significance (t-test) of the differences between the eight samples and the control group treated with PBS?
- Minor points and typos:
- page 2, row 47: “…reported to be more expressed in liver cancer…” instead of “…reported to be higher expressed in liver cancer…”;
- page 2, row 48: “…transduction pathway plays a major role…” instead of “…transduction pathway and plays a major role…”;
- page 3, row 103: “The drug loading content (DLC)…” instead of “The drug loading (DLC)…”;
- page 4, row 157: “…towards high ASPGR expressing cells.” Instead of “…towards high ASPGR expressed cells.”;
- page 5, figure 2B: it is recommended to rescaling the X-axes of the flow cytometry graphs to widen the signal peaks and make them better visible;
- page 8, row 262: how tumor inhibition rates (TIR) are calculated?
- page 10, row 326: the catalog number of commercial antibodies used in this study should be reported;
- page 10, row 328: "Protientech" instead of "Proteintech";
- page 10, rows 331-332: why were C5WN1 cells incubated in normoxia (20% O2) or hypoxia (1% O2) for 12 hours before carrying out the transwell assay experiment? This is inconsistent with the type of experiment described in the paragraph 2.2.4 and in figure 4B;
- the scale bars are missing in figures 2A, 4B, S12 and S13.
Author Response
Response to Reviewer 1
- The description of the in vitro and in vivo experiments lacks some essential information. In particular, the doses of formulations or their single components (5-FU, siRNA, etc.) are not reported in figures 4 and 6; in figure 3, only the concentrations of 5-FU (50 ug/ml) and GalNAc-pDMA (300 ug/ml) are reported in the caption and the others are missing. The concentrations of each sample, formulation or individual component, must be clearly reported for all experiments. Similarly, the treatment time is not reported for the experiment shown in figure 4A.
Response: Thanks for the suggestion. In order to easy comparison, each group use the same dose of single component and the material, now the explanations have been added in the captions. The missed concentration of siRNA and the treatment time are also added.
- How were the individual doses of formulations or single components used for the in vitro and in vivo experiments chosen? Have dose-response experiments been performed to explore a broad range of efficacy/toxicity, and ultimately to delineate the therapeutic window of these formulations? The Authors should comment on this point, since the therapeutic index is an essential factor in the evaluation of combination therapies.
Response: Thanks for the question. The therapeutic dose for in vitro experiments was determined by preliminary experiments. The text has been added in 2.2.3 and Figure S11 has been added in SI. The doses of GalNAc-pDMA and siRNA were decided by the drug loading content and siRNA N/P ratio. The therapeutic dose in in vivo experiments was selected based on the previous reported doses of 10-40 mg kg−1 (Added Ref. 36-38) and the usage amount of nanoplatform. A low dose of 5-FU at 5 mg kg−1 was used and the formulation with bimodal therapies achieved significant anti-tumor effect, which has been described and discussed in 2.3.
- The nanoparticle targeting strategy is poorly described in the Introduction and in the Scheme 1. It is necessary to better explain that targeting occurs via binding of GlcNAc to ASPGR expressed by HCC cells and that this confers selectivity with respect to normal liver tissue. Moreover, the acronym ASGPR (asialoglycoprotein receptors) should be spelled out somewhere in the text.
Response: Thanks for the suggestion. The nanoparticle targeting strategy of specific ASGPR-GalNAc recognition has been added to the introduction as recommended.
- Paragraph 2.2.1 and figure S10. The text states that water was used as a positive control of hemolysis, but this data is not shown in Figure S10.
Response: Thanks for the suggestion. The positive control was treated with water and the data was treated as 100%, which has been added to the text of Paragraph 2.2.1 and Figure S10 as recommended.
- Paragraph 2.2.2. The Authors state that the C5WN1, HepG2 and Huh7 cell lines express high levels of ASGPR on the cell membrane, while the HEK293 line used as a negative control expresses low levels of ASPGR. This claim is not supported by any data shown in this manuscript, therefore it is necessary to demonstrate different levels of ASPGR expression by immunofluorescence, western blot or cytofluorimetry.
Response: Thanks for the suggestion. It has been reported that ASGPR is specifically expressed on the cell surface of hepatic parenchymal cells (R. J. Stockert, A. G. Morell, I. H. Scheinberg, Science 1977 197,667-668, added as Ref.30) and hepatocarcinoma cells have been reported to express ASGPR abundantly (Yik JH, Saxena A, Weigel PH. J Biol Chem. 2002, 277, 23076-23083; Treichel, U.; Meyer zum Büschenfelde, K.H.; Stockert, R.J.; Poralla, T.; Gerken, G. J Gen Virol. 1994, 75, 3021-3029, added as Ref.36,37). The HEK293 cells is human embryonic kidney cell line, which is not expressed with ASGPR. The similar in vitro uptake experiments have been reported. (Ref. 11, 12, 28)
- Page 6, figure 3. Why is the PBS-treated control group (normalized to 100%) with its standard deviation bars not shown in the four graphs? What is the statistical significance (t-test) of the differences between the eight samples and the control group treated with PBS?
Response: Thanks for the question. The standard deviation of group 8 in Figure 3 is 0.09877, which is difficult displayed very clearly on the picture (shown as following enlarged part of the picture). The statistical significance of the differences between the eight samples and the PBS-treated control group has been indicated in the figure caption.
- Minor points and typos:
- page 2, row 47: “…reported to be more expressed in liver cancer…” instead of “…reported to be higher expressed in liver cancer…”;
- page 2, row 48: “…transduction pathway plays a major role…” instead of “…transduction pathway and plays a major role…”;
- page 3, row 103: “The drug loading content (DLC)…” instead of “The drug loading (DLC)…”;
- page 4, row 157: “…towards high ASPGR expressing cells.” Instead of “…towards high
- page 5, figure 2B: it is recommended to rescaling the X-axes of the flow cytometry graphs to widen the signal peaks and make them better visible;
- page 8, row 262: how tumor inhibition rates (TIR) are calculated?
- page 10, row 326: the catalog number of commercial antibodies used in this study should be reported;
- page 10, row 328: "Protientech" instead of "Proteintech";
- page 10, rows 331-332: why were C5WN1 cells incubated in normoxia (20% O2) or hypoxia (1% O2) for 12 hours before carrying out the transwell assay experiment? This is inconsistent with the type of experiment described in the paragraph 2.2.4 and in figure 4B;
- the scale bars are missing in figures 2A, 4B, S12 and S13.
Response: Thanks for the suggestions. All changes have been done according to the suggestion. The calculation of TIR has been added to Method 3.7 In vivo Antitumor Effect.
Reviewer 2 Report
Inhibition of Metastatic Hepatocarcinoma by Combined Chemotherapy with Silencing VEGF/VEGFR2 Genes through GalNAc-Modified Integrated Therapeutic System by Li et al. represents a work that is worth publishing in Molecules after considering some major issues.
- The rationale for this study should be more truly explained in the Introduction. Namely, the sentence “Herein, we synthesized an integrated delivery system based on two monomers [3-azido-2-hydroxypropyl methacrylate (AHPMA) and 2-(dimethylamino) ethyl methacrylate (DMAEMA)] to combine gene therapy against VEGF and VEGFR2 and chemotherapy to inhibit hepatocarcinogenesis and metastasis, which can be initiated with a cancer stem cell (CSC)-like cell line C5WN1 with tumorigenicity and pulmonary metastasis (Scheme 1).” is not clear. Another thing that is not well-explained: the ASGPR mediated endocytosis. This mechanism is essential for the successful entry of GalNAc into the hepatocellular carcinoma cells. The authors did not even explain the abbreviation ASGPR.
- The Results were not discussed in the context of similar or opposite findings in the literature. The Introduction cited 29 references, while the whole manuscript cited 33 references. There is no true discussion of the obtained results. The Conclusion is very short and general.
- “Hemolysis evaluation” is mentioned in the Results and Discussion (2.2.1) while there are no provided results in the manuscript and supplementary material, and no methods provided. The sentence “GalNAc-pDMA particles exhibited a very low hemolysis rate even at the highest concentration (< 10%), comparing with the serious hemolysis caused by water as a positive control (Figure S10)” is not related to the presented results in Figure S10. As I mentioned, the method of hemolysis was not described as well as water as a positive control.
- Figure Legends should provide more details. Particularly in Figure 2B where the authors only wrote “Flow cytometry analysis”.
- The authors cannot mention apoptosis because the apoptosis was not investigated: “The 5-FU/siRNA@GalNAc-pDMA treated groups of HepG2, Huh7 and C5WN1 cells displayed significant cell apoptosis with cell viabilities of 32%, 36% and 24%, respectively, while there was no effect on HEK293 cells.”
- The percentages in this sentence do not reflect the results in Figure 4A: “Surprisingly, group treated with 5-FU/siRNA@GalNAc-pDMA showed the lowest VEGF and VEGFR2 protein levels of 38% and 44%, respectively,”
- Transwell migration assay should also present a positive (FBS+) and negative control (FBS-) in order to evaluate the effects of different treatments. This methodology was not truly explained in the Experimental section. What is the relevance of this sentence in the Experimental section? “Before the experiment, C5WN1 cells were incubated in normoxia (20% O2) or hypoxia (1% O2) for 12 hours, digested with trypsin, and resuspended in serum-free medium.” Why did the authors use normoxia and hypoxia? There are no such results in the manuscript with cells treated and compared in normoxia and hypoxia.
- There is no information about the duration of treatments before protein isolation for Western blot analysis.
- In the Experimental section, the authors wrote that 6 animals per group were used, while in Figure S11, only 5 animals per group are presented.
Author Response
Inhibition of Metastatic Hepatocarcinoma by Combined Chemotherapy with Silencing VEGF/VEGFR2 Genes through GalNAc-Modified Integrated Therapeutic System
Reviewer 2
Inhibition of Metastatic Hepatocarcinoma by Combined Chemotherapy with Silencing VEGF/VEGFR2 Genes through GalNAc-Modified Integrated Therapeutic System by Li et al. represents a work that is worth publishing in Molecules after considering some major issues.
- The rationale for this study should be more truly explained in the Introduction. Namely, the sentence “Herein, we synthesized an integrated delivery system based on two monomers [3-azido-2-hydroxypropyl methacrylate (AHPMA) and 2-(dimethylamino) ethyl methacrylate (DMAEMA)] to combine gene therapy against VEGF and VEGFR2 and chemotherapy to inhibit hepatocarcinogenesis and metastasis, which can be initiated with a cancer stem cell (CSC)-like cell line C5WN1 with tumorigenicity and pulmonary metastasis (Scheme 1).” is not clear. Another thing that is not well-explained: the ASGPR mediated endocytosis. This mechanism is essential for the successful entry of GalNAc into the hepatocellular carcinoma cells. The authors did not even explain the abbreviation ASGPR.
Response: Thanks for the suggestions. The description of rationale design of the platform and the ASGPR-mediated mechanism have been added in the introduction part.
- The Results were not discussed in the context of similar or opposite findings in the literature. The Introduction cited 29 references, while the whole manuscript cited 33 references. There is no true discussion of the obtained results. The Conclusion is very short and general.
Response: Thanks for the suggestions. In the results and discussion 2.1, there are ref. 30-33. The discussion and conclusion part has been modified.
- “Hemolysis evaluation” is mentioned in the Results and Discussion (2.2.1) while there are no provided results in the manuscript and supplementary material, and no methods provided. The sentence “GalNAc-pDMA particles exhibited a very low hemolysis rate even at the highest concentration (< 10%), comparing with the serious hemolysis caused by water as a positive control (Figure S10)” is not related to the presented results in Figure S10. As I mentioned, the method of hemolysis was not described as well as water as a positive control.
Response: Thanks for the suggestion. The data of positive control for the hemolysis assay has been added to Figure S10 and the positive and negative control has been described in the Supporting Information.
- Figure Legends should provide more details. Particularly in Figure 2B where the authors only wrote “Flow cytometry analysis”.
Response: Thanks for the suggestion. All Figure legends have been added details.
- The authors cannot mention apoptosis because the apoptosis was not investigated: “The 5-FU/siRNA@GalNAc-pDMA treated groups of HepG2, Huh7 and C5WN1 cells displayed significant cell apoptosis with cell viabilities of 32%, 36% and 24%, respectively, while there was no effect on HEK293 cells.”
Response: Thanks for the suggestion. “significant cell apoptosis” has been modified to “significant cell death”.
- Th percentages in this sentence do not reflect the results in Figure 4A: “Surprisingly, group treated with 5-FU/siRNA@GalNAc-pDMA showed the lowest VEGF and VEGFR2 protein levels of 38% and 44%, respectively,”
Response: Thanks for the question. It has been corrected.
- Transwell migration assay should also present a positive (FBS+) and negative control (FBS-) in order to evaluate the effects of different treatments. This methodology was not truly explained in the Experimental section. What is the relevance of this sentence in the Experimental section? “Before the experiment, C5WN1 cells were incubated in normoxia (20% O2) or hypoxia (1% O2) for 12 hours, digested with trypsin, and resuspended in serum-free medium.” Why did the authors use normoxia and hypoxia? There are no such results in the manuscript with cells treated and compared in normoxia and hypoxia.
Response: Thanks for the question. We didn’t use normoxia and hypoxia, which is a careless mistake. The method part has been corrected. The addition of FBS as a chemoattractant is an essential step in migration experiments. In this study, the effect of 5-FU/siRNAGalNAc-pDMA on cell migration was examined and compared with the other groups with different treatments.
- There is no information about the duration of treatments before protein isolation for Western blot analysis.
Response: Thanks for the question. It has been added in the legend of Figure 4 and the text of method.
- In the Experimental section, the authors wrote that 6 animals per group were used, while in Figure S11, only 5 animals per group are presented.
Response: Thanks for the question. It has been corrected.
Round 2
Reviewer 1 Report
All points raised have been satisfactorily addressed, therefore publication of the manuscript in this form is recommended.